# A novel chimeric RNA originating from BmCPV S4 and *Bombyx mori* HDAC11 transcripts regulates virus proliferation

**Jun Pan**[1☉], **Shulin Wei**[1☉], **Qunnan Qiu**[1☉], **Xinyu Tong**[1], **Zeen Shen**[1,2], **Min Zhu**[1], **Xiaolong Hu**[1,2], **Chengliang Gong**[1,2]*

1 School of Biology and Basic Medical Sciences, Soochow University, Suzhou, Jiangsu, People's Republic of China, 2 Agricultural Biotechnology Research Institute, Agricultural biotechnology and Ecological Research Institute, Soochow University, Suzhou, People's Republic of China

☉ These authors contributed equally to this work.
* gongcl@suda.edu.cn

**Data Availability Statement:** All relevant data are within the manuscript and its Supporting Information files. What's more, the raw RNA_Seq data have been uploaded to the NCBI database

## Abstract

Polymerases encoded by segmented negative-strand RNA viruses cleave 5'-m7G-capped host transcripts to prime viral mRNA synthesis ("cap-snatching") to generate chimeric RNA, and trans-splicing occurs between viral and cellular transcripts. *Bombyx mori* cytoplasmic polyhedrosis virus (BmCPV), an RNA virus belonging to Reoviridae, is a major pathogen of silkworm (*B. mori*). The genome of BmCPV consists of 10 segmented double-stranded RNAs (S1–S10) from which viral RNAs encoding a protein are transcribed. In this study, chimeric silkworm-BmCPV RNAs, in which the sequence derived from the silkworm transcript could fuse with both the 5' end and the 3' end of viral RNA, were identified in the midgut of BmCPV-infected silkworms by RNA_seq and further confirmed by RT-PCR and Sanger sequencing. A novel chimeric RNA, HDAC11-S4 RNA 4, derived from silkworm histone deacetylase 11 (HDAC11) and the BmCPV S4 transcript encoding viral structural protein 4 (VP4), was selected for validation by *in situ* hybridization and Northern blotting. Interestingly, our results indicated that HDAC11-S4 RNA 4 was generated in a BmCPV RNA-dependent RNA polymerase (RdRp)-independent manner and could be translated into a truncated BmCPV VP4 with a silkworm HDAC11-derived N-terminal extension. Moreover, it was confirmed that HDAC11-S4 RNA 4 inhibited BmCPV proliferation, decreased the level of H3K9me3 and increased the level of H3K9ac. These results indicated that during infection with BmCPV, a novel mechanism, different from that described in previous reports, allows the genesis of chimeric silkworm-BmCPV RNAs with biological functions.

## Author summary

It has previously been reported that start codons with cap-snatched host transcripts or trans-splicing can generate chimeric RNAs on ssRNA and dsDNA viruses; however, whether chimeric RNAs can be generated on dsRNA viruses remains a mystery. We identified host-virus chimeric RNAs in a dsRNA virus infection model (BmCPV in silkworm),

(accession numbers SRR22891215, SRR22891214, SRR22891213) and the data that support the findings of this study are publicly available from bioRxiv with the identifier(s) https://doi.org/10.1101/2023.02.07.527451.

**Funding:** This research was supported by the National Natural Science Foundation of China (32372946G, 32072792G, 31872424G, and 31972620H) and the Priority Academic Program of Development of Jiangsu Higher Education Institutions (YL13430023, for the school). The funders had no role in the study design, data collection, analysis, decision to publish, or manuscript preparation.

**Competing interests:** The authors have declared that no competing interests exist.

which is a novel phenomenon. What's more, our results indicated that the selected chimeric RNA HDAC11-S4 RNA 4 (derived from silkworm histone deacetylase 11 (HDAC11) and the BmCPV S4 transcript encoding viral structural protein 4 (VP4)) was generated in a BmCPV RNA-dependent RNA polymerase (RdRp)-independent manner and could be translated into a truncated BmCPV VP4 with a silkworm HDAC11-derived N-terminal extension. Further, We found that the chimeric RNA inhibited virus proliferation and decreased histone methylation, and increased histone acetylation. That is to say, the chimeric protein has activity and provides novel functionality compared to either of the normal versions of the two components on their own.

## 1 Introduction

It has been known that two distinct RNA transcripts can be joined by trans-splicing to form chimeric RNA [1]. A previous study showed naturally occurring heterologous trans-splicing of adenovirus (ADV) (linear double-stranded DNA virus) RNA with host cellular transcripts during infection [2]. Moreover, it has been found that heterologous trans-splicing also occurs between human immunodeficiency virus (HIV, single-stranded RNA virus)-nef RNA and cellular transcripts [3], and a 100 kDa super T antigen harboring a duplication of the retinoblastoma (pRb)-binding domain can be generated by homologous RNA trans-splicing of viral early transcripts during infection with simian virus 40 (SV40, cyclic double-stranded DNA virus) [4]. Chimeric RNAs can also be generated by heterologous SV40 transcript trans-splicing, but their function is unknown [5]. A recent study showed that novel proteins associated with the viral life cycle were produced by trans-splicing of late transcripts of human polyoma JC virus (JCV, cyclic double-stranded DNA virus) [6].

Moreover, it has been clarified that segmented negative-strand RNA viruses (sNSVs) such as Peribunyaviridae, Phenuiviridae, Tospoviridae, Arenaviridae, Nairoviridae, Hantaviridae, and Orthomyxoviridae families cannot synthesize 5' methyl-7-guanosine ($m^7G$) cap structure for their mRNAs [7,8,9], but can generate viral mRNA with an $m^7G$ cap structure through a "cap-snatching" mechanism, which initiates the synthesis of viral mRNA using the capped primer formed by cleaving the capped host transcript [10]. The 5'-end of chimeric host and virus mRNAs generated via the cap-snatching mechanism have highly diversified sequences derived from the 5'-ends of host mRNAs [11,12]. Unlike canonical cap-snatching, the influenza A virus (IAV) can utilize noncanonical cap-snatching to diversify its mRNAs/ncRNAs [13]. It has been reported that sNSVs can obtain functional upstream start codons (uAUGs) within cap-snatched host transcripts, via a process termed "start-snatching" [8]. Previous studies suggested that novel viral proteins generated by start-snatching impact viral replication and serve as additional targets for host surveillance [8,14].

In addition to trans-splicing and cap-snatching, it has been reported that a functional chimeric mRNA encoding a fusion of the viral E3 ubiquitin ligase ICP0 and viral membrane glycoprotein L can be produced by low level readthrough transcription during herpes simplex virus type 1 (HSV-1) infection [15] and that chimeric cellular-HIV mRNAs can be generated by aberrant splicing [16].

Cytoplasmic polyhedrosis viruses (CPVs) with segmented double-stranded RNAs (dsRNAs) that are packaged into a single-layered icosahedral viral capsid [17] belong to the *Cypovirus* genus of the Reoviridae family. CPVs can infect insects belonging to Lepidoptera, Hymenoptera and Coleoptera and play a very important role in the control of pest populations in agriculture and forestry. *Bombyx mori* CPV (BmCPV), a model CPV species, is a pathogen

of silkworm, *B. mori*, and specifically infects epithelial cells of the midgut, resulting in a decrease in cocoon production [18]. The BmCPV genome consists of 10 dsRNA segments [19]. Previous studies have indicated that the viral structural proteins VP1, VP2, VP3, VP4, VP6 and VP7 are encoded by the S1, S2, S3, S4, S6 and S7 segments, respectively, and that the nonstructural proteins NSP5, NSP8, NSP9 and polyhedrin are encoded by the S5, S8, S9 and S10 segments [20]. The transcript of each segment possesses a 5'-m7G cap structure [21], which is considered a monocistron encoding a protein. In recent decades, it was widely believed that viral transcripts were not spliced during the formation of mature viral mRNAs. However, recent studies have indicated that BmCPV RNAs can be cut by Dicer-2 and an uncharacterized endo-RNase to produce thousands of small viral RNAs (vsRNAs) as a strategy for opposing virus infection [22], and transcripts of BmCPV can also be processed to viral microRNAs (vmiRNAs) to promote virus infection [23,24,25,26]. Our previous studies indicated that BmCPV RNAs can be processed to form viral circular RNAs (vcircRNAs) with an unknown mechanism. Both circRNA-vSP27 and cirRNA 000048, derived from BmCPV, can attenuate viral replication because they encode the small peptides vSP27 and vsp21 [27,28], respectively. Moreover, a BmCPV S7 RNA-derived circular DNA (vcircDNA) referred to as vcDNA-S7 was detected in infected cells, and vcDNA-S7 was transcribed into RNA, which was further processed into antiviral vsRNAs [29]. These results indicated that functional molecules, including vsRNAs, vmiRNAs, vcircRNAs and vcircDNA, can be formed from BmCPV RNAs. However, whether chimeric silkworm-BmCPV RNAs can be generated during BmCPV infection remains a mystery.

In this study, we identified chimeric silkworm-BmCPV RNAs and validated a novel chimeric RNA, HDAC11-S4 RNA 4, derived from silkworm histone deacetylase 11 (HDAC11), and BmCPV S4 RNA transcripts were shown to be generated in a BmCPV RNA-dependent RNA polymerase (RdRp)-independent manner. Although chimeric host-virus RNA formation using a cap-snatching mechanism has been previously reported for many different host-virus infections, our work showed and characterized for the first time the occurrence of chimeric silkworm-BmCPV RNA possibly generated using a different mechanism. Furthermore, we attempted to characterize the biological effect of this HDAC11-S4 RNA 4 chimeric RNA in virus-infected BmN cells, and found that a truncated BmCPV VP4 with a silkworm HDAC11-derived N-terminal extension, translated by HDAC11-S4 RNA 4, could inhibit BmCPV proliferation, decrease the level of H3K9me3 and increase the level of H3K9ac. These results indicated that a novel mechanism, different from that described in previous reports, allows the genesis of chimeric silkworm-BmCPV RNAs with biological functions.

## 2 Results

### 2.1 Chimeric silkworm-BmCPV RNAs can be formed during BmCPV infection

It has been found that chimeric host–virus RNAs can be formed by cap-snatching during sNSV and HIV infection [3,8] or by trans-splicing during dsDNA virus infection [4,6]; however, chimeric host-viral RNAs have not been found during dsRNA virus infection thus far. To explore whether chimeric silkworm-BmCPV RNA can be formed, the BmCPV-infected midgut was subjected to RNA_Seq. The raw sequencing data have been uploaded to the NCBI database (accession numbers SRR22891215, SRR22891214, SRR22891213). After removing low-quality reads, the clean reads were assembled with StringTie software (1.2.0) [30] using the *B. mori* genome and the BmCPV genome as references. STAR-Fusion was used to analyze fusion transcripts [31]. A total of 516 redundant chimeric silkworm RNAs were identified, and changes in the number and abundance of chimeric transcripts were found following virus

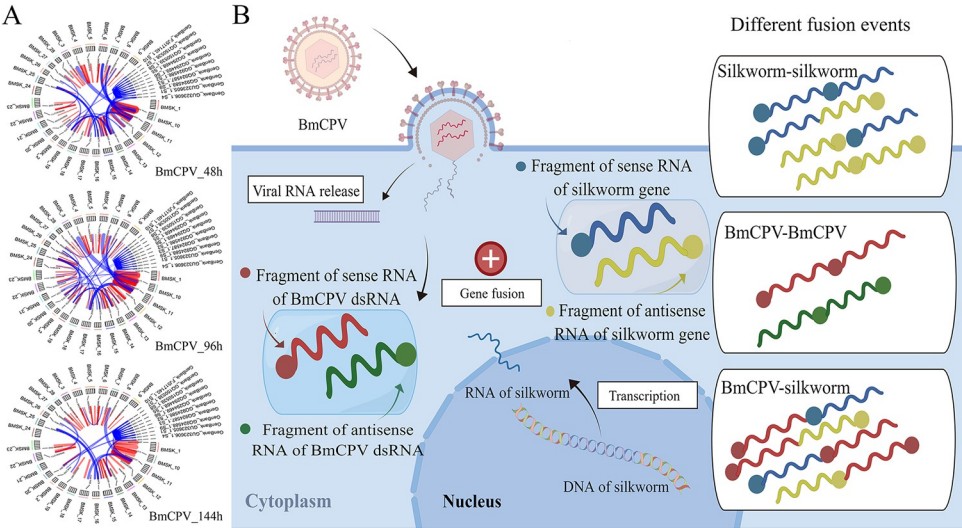

**Fig 1. Chimeric RNAs generated in the silkworm midgut infected with BmCPV. A**, Circos diagram of chimeric RNAs in the midgut at 48 h (BmCPV_48), 96 h (BmCPV_96) and 144 h (BmCPV_144) post-infection with BmCPV; **B**, Fusion types of chimeric RNAs detected in the midgut infected with BmCPV. "BMSK_X" indicates the individual silkworm chromosomes. Each line in the Circos plots represents a fusion event; the red line represents the fusion event between two RNA molecules originating from the same chromosome, while the blue line represents the fusion event between RNA molecules originating from different chromosomes/BmCPV genomic dsRNA segment. The circle at one end of the wavy line represents the 5'-end of RNA molecule.

infection. Interestingly, 372 redundant chimeric viral RNAs and 34 redundant chimeric silkworm-viral RNAs (S1 Table) were found in the BmCPV-infected midgut (Fig 1). Most of the chimeric silkworm-silkworm RNAs were derived from two different transcripts, but some chimeric silkworm-silkworm RNAs were formed by the fusion of 3 fragments from different transcripts (Fig 1). Among the chimeric virus–virus RNAs, all detected chimeric RNAs originated from fusions between different BmCPV RNA fragments, which were derived from either the fusion of sense strands of viral RNA fragments or the fusion of antisense strands of viral RNA fragments (Fig 1). Among the chimeric silkworm-BmCPV RNAs, chimeric RNAs derived from the fusion of sense/antisense RNA fragments of silkworm genes and sense/antisense RNA fragments of BmCPV genomic dsRNAs were identified in the BmCPV-infected midgut. Silkworm RNA sequences can be fused with the 5' or 3' terminus of a viral RNA fragment (Fig 1). Among the chimeric RNAs, the most frequently detected RNA fragment derived from silkworm transcripts was large subunit ribosomal RNA (XR_005246581.1), followed by the mRNA clone fcaL43P13 (AK384927.1), U6 atac minor spliceosomal RNA (XR_005245887.1), Bm_160 RNA (GU247410.1) and histone deacetylase (HDAC) 11 (XM_004925365.4). The flanking sequences of the junction sites of the selected chimeric RNAs were identified by RT-PCR and Sanger sequencing, and the results were consistent with those of high-throughput sequencing, suggesting that the results obtained from high-throughput sequencing were credible (Fig 2).

## 2.2 Chimeric silkworm-BmCPV RNAs are formed in a different way than previously reported

The trans-splicing of viral transcripts with host or viral transcripts can occur in the nucleus of infected cells through a splicing mechanism [3,4,5,6,16]. In this study, we defined the parental RNAs of the sequences located upstream/downstream of the fusion site of chimeric silkworm-

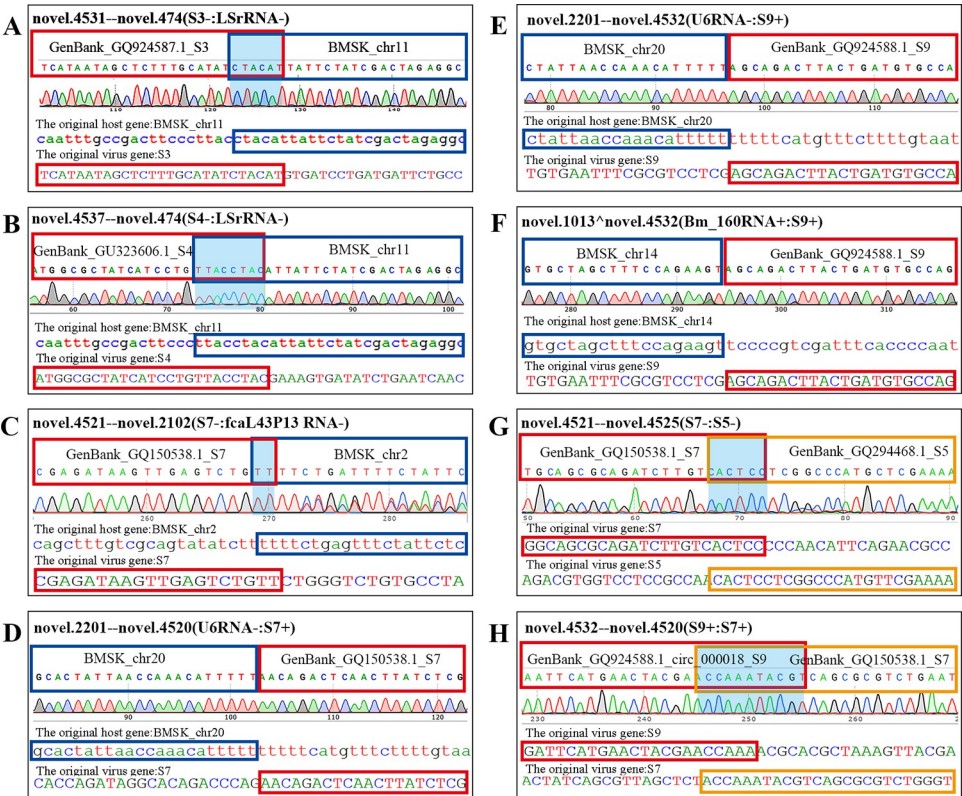

**Fig 2. Identification of chimeric RNAs by RT-PCR and Sanger sequencing. A and B**, fragments from the antisense RNAs of BmCPV S3 (A) and S4 (B) dsRNAs fused with a fragment of the larger subunit of ribosomal RNA (LSrRNA) from silkworm, the sequence indicated with a blue background is a common sequence shared by the S3/S4 and LSrRNA antisense sequences. **C**, Fragment of antisense RNA of BmCPV S7 dsRNA fused with a fragment of the silkworm mRNA clone: fcaL43P13. **D**, Fragment of silkworm U6 ncRNA fused with a fragment of sense RNA BmCPV S7 dsRNA. **E and F**, a fragment of silkworm U6 ncRNA/Bm_160 RNA fused with a fragment of the sense RNA of BmCPV S9 dsRNA. **G**, Fragment of antisense RNA of BmCPV S7 dsRNAs fused with a fragment of the antisense RNA of BmCPV S5 dsRNAs. **H**, A fragment of the sense RNA of BmCPV S9 dsRNAs fused with a fragment of the sense RNA of BmCPV S7 dsRNAs. The sequences indicated with a blue background are sequences shared by two parental RNAs.

BmCPV RNAs as left/right RNAs. To understand the formation mechanisms of chimeric RNAs, we analyzed the flanking sequences of the breakpoints of the parental RNAs, and the conserved sequences required for the splicing mechanism were not found (S1 Fig). The transcription of BmCPV RNAs occurred in the cytoplasm, suggesting that chimeric silkworm-BmCPV RNAs were formed in a manner different from trans-splicing.

Cap-snatching is a mechanism applied by sNSVs to initiate genome transcription [32]. The 5'-ends of generated chimeric host and virus mRNAs have highly diversified sequences derived from the 5'-ends of host mRNAs [11,12]. In this study, chimeric silkworm-BmCPV RNAs were found to be formed by the fusion of sense/antisense RNA fragments of silkworm genes and sense/antisense RNA fragments of BmCPV genomic RNAs. A silkworm RNA sequence can be fused with the 5' or 3' terminus of a viral RNA fragment, implying that chimeric silkworm-BmCPV RNAs are formed in a manner different from cap-snatching (Fig 1).

Moreover, the fusion events between silkworm and BmCPV RNAs can be divided into two types. In one of these categories, the 5'-flanking sequence of the left parental RNA breakpoint and the 3'-flanking sequence of the right parental RNA breakpoint share a common sequence of 1–10 nt, and only one common sequence is retained in the formed chimeric RNA. In the

other, two RNA fragments derived from different RNAs are directly joined after the RNAs are broken (Fig 2).

## 2.3 Isoforms of chimeric silkworm HDAC11-S4 RNA truly exist in the midgut infected with BmCPV

In this study, chimeric silkworm HDAC11-S4 RNA was identified in the BmCPV-infected midgut by high-throughput sequencing. To confirm this result, RT–PCR was carried out with two pairs of primers (HDAC11+:S4 RNA+)20-123 and (HDAC11+:S4 RNA+)20-456 (S2 Table), which were designed according to the flanking sequences of the junction site of chimeric silkworm HDAC 11-S4 RNA. The sequencing results for the RT-PCR products showed that 6 isoforms (HDAC11-S4 RNA 1–6) of HDAC 11-S4 RNA were identifiable, suggesting that the number of chimeric silkworm-BmCPV RNAs obtained by RNA_Seq was underestimated. With the exception of HDAC11-S4 RNA 2, the right flanking sequences of the junction sites of the remaining chimeric RNAs were from the 884–3262 nt region of the sense chain of S4 dsRNA (GU323606.1) (Fig 3). Moreover, the sequencing results showed that the base at position 895 of S4 RNA was 'T' (Fig 3D), but there is also case where it is 'A'(S2 Fig). This result may be related to mutation (T895A) in the S4 RNA sequences of some viruses. T895A mutation does not affect the amino acid sequence of VP4 protein, because this point mutation occurs at the third base of the codon.

To eliminate misjudgment caused by sequencing artifacts, reverse transcriptional noise, template-switching and scrambled junctions of RNAs, HDAC11-S4 RNA 4 was selected for further validation. RT-PCR was conducted with the adapter HCPV-20-4 primer pair, in which one primer targeted the junction site of HDAC11-S4 RNA 4, and the other targeted the downstream sequence of the junction site of HDAC11-S4 RNA 4. The sequence of the RT-PCR product was consistent with the expected result. The total RNAs extracted from the midgut infected with BmCPV were identified by Northern blotting with a biotin-labeled probe targeting the junction site of HDAC11-S4 RNA 4, and the results showed that a specific signal band

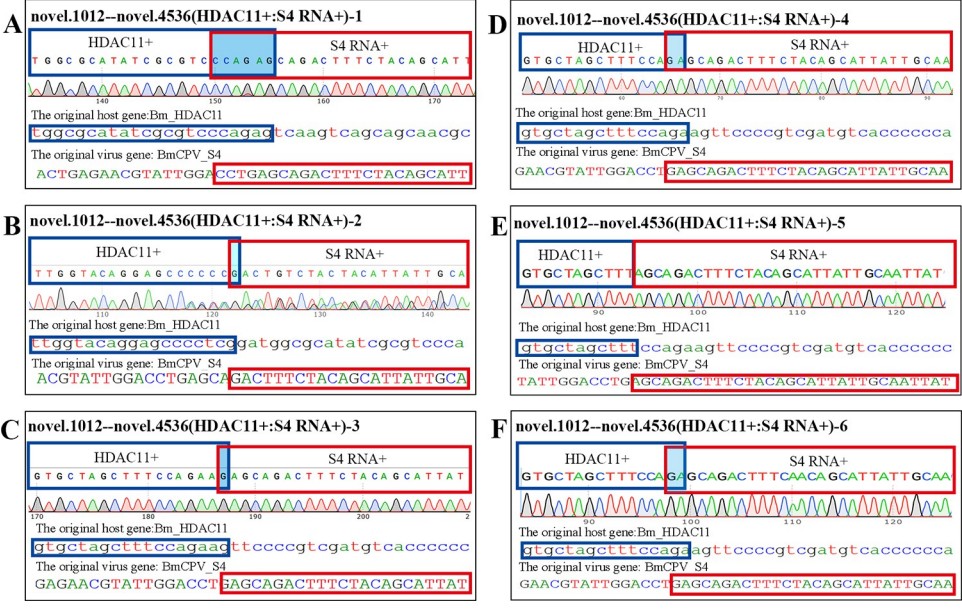

**Fig 3. Isoforms of chimeric silkworm HDAC11-S4 RNA.** A, B, C, D, E and F represent the junction sites of isoforms of chimeric silkworm HDAC11-S4 RNA.

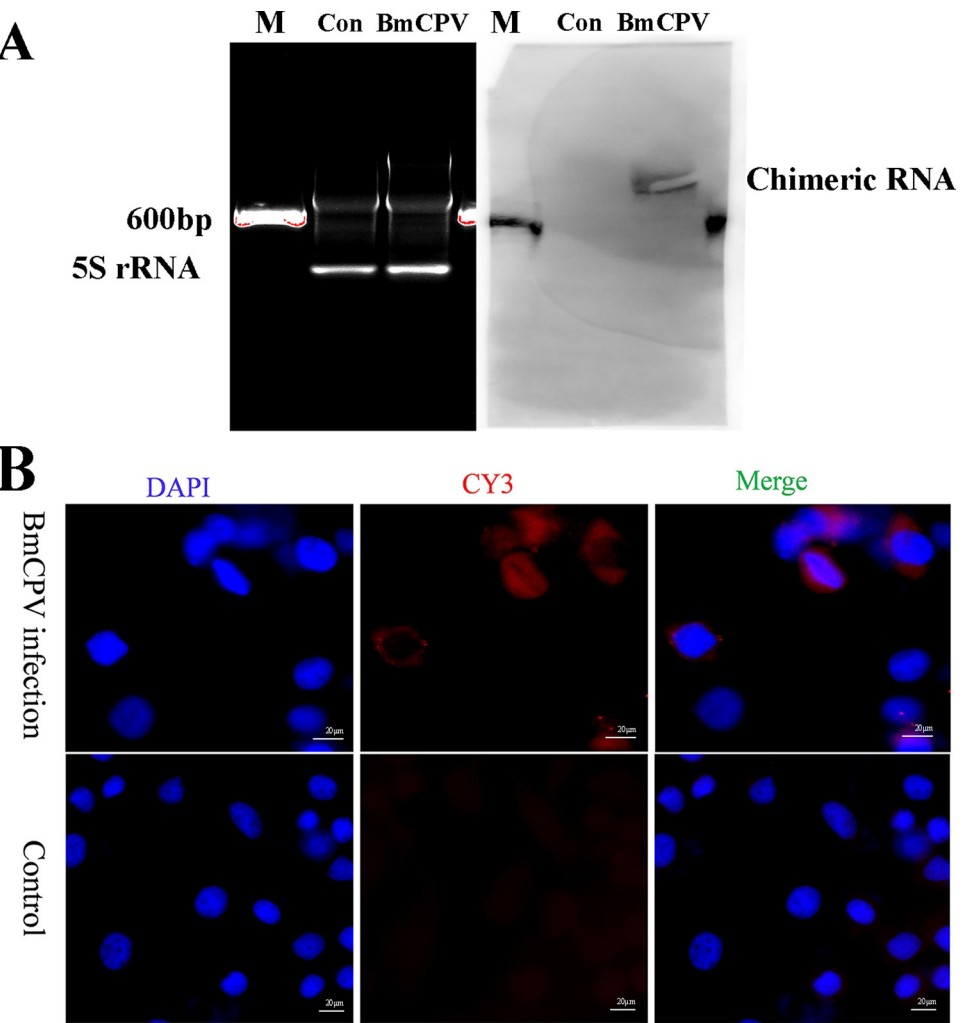

**Fig 4. HDAC11-S4 RNA 4 was validated by Northern blotting and in situ hybridization. A,** Validation of HDAC11-S4 RNA 4 by Northern blotting. The total RNA extracted from the BmCPV-infected midgut was separated on 1% agarose-formaldehyde gels (left) and transferred to Hybond-N+ nylon membranes. Northern blotting was conducted with biotin-labeled DNA targeting the junction site of chimeric HDAC11-S4 RNA 4 (Right). Total RNAs extracted from the non-BmCPV-infected midgut was used as a control. Lane M, DNA marker; Lane Con, total RNAs extracted from the non-BmCPV-infected midgut; Lane BmCPV, total RNAs extracted from the BmCPV-infected midgut. **B,** Validation of HDAC11-S4 RNA 4 by *in situ* hybridization. BmN cells with/without BmCPV infection at 48 h post-infection were digested with proteinase K and hybridized with biotin-labeled DNA targeting the junction site of chimeric HDAC11-S4 RNA 4. Hybridization signals were detected using CY3-labeled streptavidin. Cell nuclei were counterstained with DAPI.

was found in the RNAs extracted from the midgut infected with BmCPV (Fig 4A). Moreover, BmCPV-infected BmN cells were detected by *in situ* hybridization with the probe mentioned above, and as expected, HDAC11-S4 RNA 4 was identified in the cytoplasm of infected cells (Fig 4B). To confirm that the probe is binding the junction site and not HDAC11 RNA/virus S4 RNA, the prepared HDAC11-S4 RNA 4, HDAC11 RNA, and S4 RNA by *in vitro* transcription were separated by electrophoresis, and then subjected to Southern hybridization using the probe. Hybridization signal was only detected in HDAC11-S4 RNA sample (S3 Fig). Similarly, when *in situ* hybridization was performed on *Ctenopharyngodon idellus* kidney (CIK) cells respectively transfected with HDAC11-S4 RNA 4, HDAC11 RNA, and S4 RNA, hybridization signals were only detected in the cells transfected with HDAC11-S4 RNA 4 (S3 Fig). These

results demonstrated that the hybridization signal comes from the binding of the probe to the junction site, rather than the host HDAC11 RNA or the viral S4 RNA.

## 2.4 Chimeric HDAC11-S4 RNA 4 is formed in a BmCPV-encoded RdRp-independent manner

The RNA polymerase encoded by sNSVs is required for cap-snatching [10]. To determine whether RdRp encoded by BmCPV is required for the formation of chimeric HDAC11-S4 RNA 4, pIZT-CS4 with the complete cDNA sequence of BmCPV S4 dsRNA [33] and a mixture of pIZT-CS4 and pIZT-CS2 with the complete cDNA sequence of BmCPV S2 dsRNA encoding RdRp were transfected into BmN cells, and the total RNAs extracted at 48 h posttransfection were used to evaluate the formation of chimeric HDAC11-S4 RNA 4 by RT-PCR. The desired specific PCR product could be amplified from the two extracted RNA samples, but the specific PCR product could not be obtained from the RNAs extracted from the BmN cells without transfection, suggesting that the formation of chimeric HDAC11-S4 RNA 4 is not mediated by RdRp which is encoded by BmCPV (Fig 5A).

## 2.5 The level of chimeric HDAC11-S4 RNA 4 increases upon viral infection

To understand the HDAC11-S4 RNA 4 expression pattern, qRT-PCR was conducted. The results showed that the level of chimeric HDAC11-S4 RNA 4 increased following BmCPV infection, but the expression level decreased at 144 h post-infection (Fig 5B). Moreover, the ratio of HDAC11-S4 RNA 4, S4 RNA, and HDAC11 RNA in BmCPV-infected BmN cells at 48 h post-infection was estimated using RT-qPCR, and the results showed that HDAC11-S4 RNA4: S4 RNA: HDAC11 was 1:19:139, indicating that only a fraction of S4 RNA and HDAC11 RNA become chimeric RNAs.

## 2.6 BmCPV replication is regulated by chimeric HDAC11-S4 RNA 4

In this study, 6 isoforms (HDAC11-S4 RNA 1–6) of HDAC 11-S4 RNA were identified, in which, HDAC11-S4 RNA 4 may have the potential to encode a truncated viral VP4 with N-

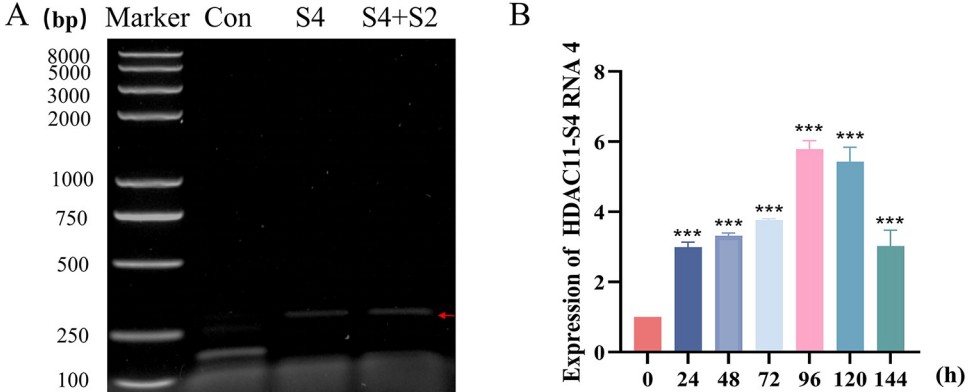

**Fig 5. Formation and expression pattern of chimeric HDAC11-S4 RNA 4. A,** Formation of chimeric HDAC11-S4 RNA 4. pIZT-CS4 and a mixture of pIZT-CS4 and pIZT-CS2 were transfected into BmN cells, and total RNAs were extracted at 48 h posttransfection to determine the formation of chimeric HDAC11-S4 RNA 4 with RT-PCR. **B,** Expression pattern of chimeric HDAC11-S4 RNA 4. The total RNAs extracted from the midgut infected with BmCPV at 0–144 h postinfection were used as templates, and the expression level of chimeric HDAC11-S4 RNA 4 was determined by qRT–PCR. The TIF-4A gene was used as an internal reference. Each biological experiment was repeated three times. ***, p<0.001.

terminal extensions derived from host HDAC11 RNA (see below), therefore, HDAC11-S4 RNA 4 was chosen for further research. To understand the function of chimeric HDAC11-S4 RNA 4, the effect of transfection with different doses of chimeric HDAC11-S4 RNA 4 on BmCPV replication in the cultured BmN cells was investigated. Meanwhile, the transfection with the corresponding doses of partial HDAC11 RNA (the upstream sequence of the junction site of chimeric HDAC11 S4 RNA 4), CPV S4 RNA, and green fluorescent protein (GFP) RNA were used as controls respectively (Fig 6A). The qRT–PCR results showed that the *vp1* gene expression level decreased in chimeric HDAC11-S4 RNA 4-transfected BmN cells in a transfection dose-dependent manner (Fig 6F–6H). Western blot results indicated that VP7 expression was also decreased in chimeric HDAC11-S4 RNA 4-transfected BmN cells compared to GFP RNA-transfected BmN cells (Fig 6B–6C), suggesting that BmCPV replication was inhibited. Moreover, compared to partial HDAC11 RNA- and viral S4 RNA-transfected BmN cells, the VP7 protein level and the *vp1* transcription level decreased in the transfected BmN cells with chimeric HDAC11-S4 RNA 4.

## 2.7 H3K9me3/H3K9ac of histone 3 is regulated by chimeric HDAC11-S4 RNA 4

Chimeric HDAC11-S4 RNA 4 was a hybrid molecule consisting of the HDAC11 gene transcript and a fragment of the sense RNA of BmCPV S4 dsRNA. Therefore, the effect of chimeric HDAC11-S4 RNA 4 on the methylation and acetylation of histone H3 was determined by Western blotting. The results showed that the trimethylation of histone 3 lysine 9 (H3K9me3) was decreased and the acetylation of histone 3 lysine 9 (H3K9ac) was increased in the chimeric HDAC11-S4 RNA 4-transfected BmN cells compared with BmN cells transfected with GFP RNA (Fig 6B, 6D, 6E). Furthermore, the level of H3K9me3 decreased, the level of H3K9ac increased in the transfected BmN cells with the chimeric HDAC11-S4 RNA 4, compared to the transfected BmN cells with partial HDAC11 RNA or viral S4 RNA, suggesting that the chimeric HDAC11-S4 RNA 4 provides novel functionality compared to either of the normal versions of the two components on their own.

## 2.8 Chimeric HDAC11-S4 RNA 4 encodes a truncated viral VP4 with N-terminal extensions derived from host HDAC11 RNA

To explore whether chimeric HDAC11-S4 RNA 4 has the potential to encode proteins, ORF finder (https://www.ncbi.nlm.nih.gov/orffinder/) was used to predict the open reading frame (ORF) of the chimeric HDAC11-S4 RNA 4. The results showed that the chimeric RNA may encode a novel protein (termed NesVP4) which is a truncated viral VP4 (768 amino acid residues) with N-terminal extensions (20 amino acid residues) derived from host HDAC11 RNA (Fig 7A and 7B). To confirm this result, the BmCPV-infected midgut was analyzed in different infection phases by Western blotting with an anti-VP4 antibody, and two specific signal bands representing VP4 (116 kDa) and NesVP4 (87 kDa) were observed, implying that NesVP4 was translated by the chimeric RNA (Fig 7C). Moreover, the specific signal band representing NesVP4 could be observed in the cells transfected with HDAC11-S4 RNA 4 obtained by *in vitro* transcription (Fig 7D). These results indicated that NesVP4 was translated by HDAC11-S4 RNA 4. Besides, the function of chimeric RNAs *in vivo* is closely related to its abundance. The junction read count obtained by high-throughput sequencing (S1 Table) suggested that the abundance of HDAC 11-S4 RNA was not high. The relative abundance of HDAC11-S4 RNA 4 was estimated using qRT-PCR, the results showed that the abundance of HDAC11-S4 RNA 4 was approximately 1/19 of that of S4 RNA in BmN cells infected with BmCPV for 48 h. Furthermore, the abundance of HDAC11-S4 RNA 4 increases with virus

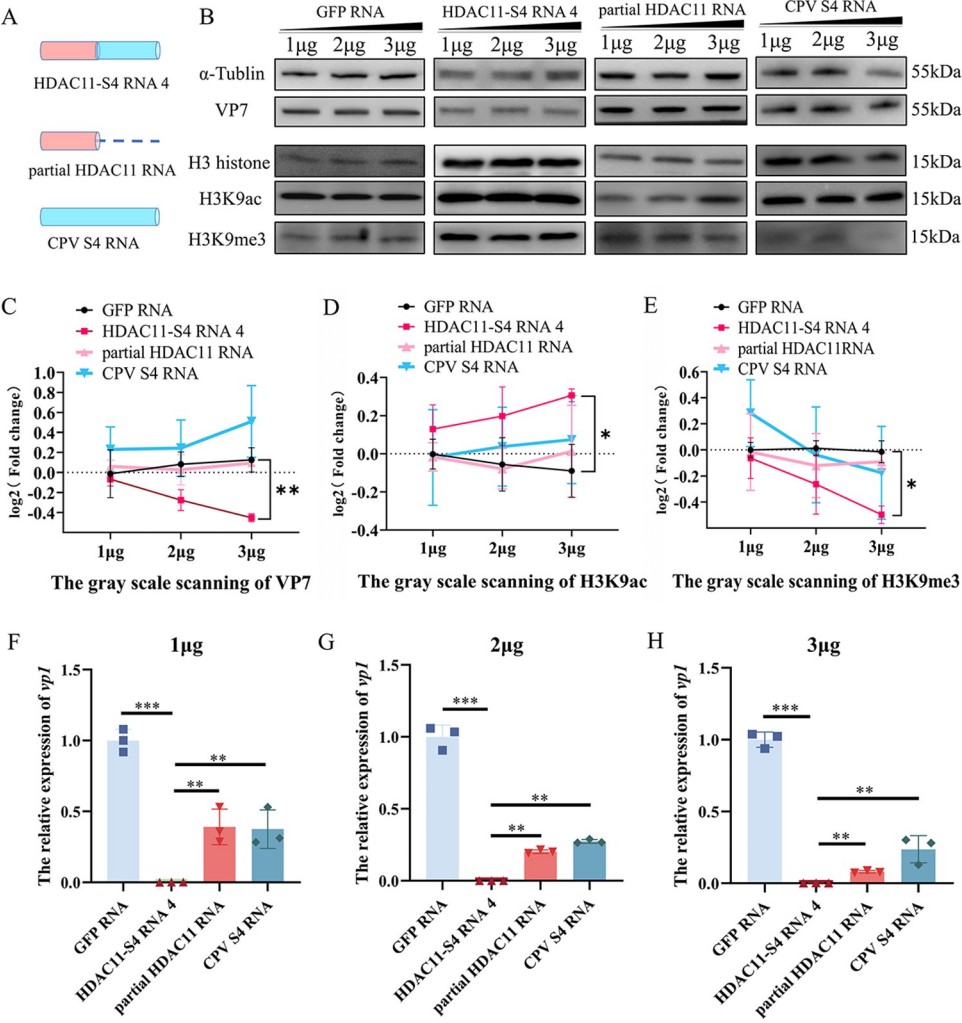

**Fig 6. Effect of chimeric HDAC11-S4 RNA 4 on BmCPV viral gene expression and H3K9me3/H3K9ac of histone.**
**A**, Schematic diagram of various RNA molecules in the chimeric HDAC11-S4 RNA 4, HDAC11-S4 RNA 4, a chimeric RNA derived from HDAC11 RNA and the BmCPV S4 RNA; partial HDAC11 RNA, the upstream sequence of the junction site of chimeric HDAC11 S4 RNA 4; CPV S4 RNA, the RNA derived from BmCPV segment S4 RNA. **B**, Effect of chimeric HDAC11-S4 RNA 4 on BmCPV VP7 and H3K9me3/H3K9ac of histone 3 levels. BmN cells ($10^6$) transfected with 1, 2 or 3 μg of chimeric HDAC11-S4 RNA 4 were inoculated with BmCPV at 24 h post-transfection, and the VP7 protein level and H3K9me3/H3K9ac of histone 3 were determined at 48 h post-infection by Western blotting with anti-VP7, anti-H3K9me3 and anti-H3K9ac antibodies. The transfection with the corresponding doses of GFP RNA, partial HDAC11 RNA and CPV S4 RNA (produced by *in vitro* transcription) were used as controls, respectively. **C, D, E,** The grayscale intensity of the Western blot signaling bands in Fig 6B were analyzed by Image J software. F, G, H, Effect of chimeric HDAC11-S4 RNA 4 on BmCPV *vp1* gene expression levels. BmN cells ($10^6$) transfected with 1, 2 or 3 μg of chimeric HDAC11-S4 RNA 4 were inoculated with BmCPV at 24 h posttransfection, and the relative expression level of the BmCPV *vp1* gene was determined at 48 h posttransfection by qRT–PCR. GFP RNA, partial HDAC11 RNA and CPV S4 RNA-transfected cells infected with BmCPV were used as controls. Each biological experiment was repeated three times. **, $p<0.01$; *, $p<0.05$.

proliferation (Fig 5), suggesting that the NesVP4 chimeric protein encoded by HDAC11-S4 RNA 4 is potentially biologically relevant.

## 3 Discussion

Previous studies have indicated that chimeric RNAs can be generated during the infection of sNSVs, HIV (single-stranded RNA virus), and SV40/JCV/HSV-1/ADV (double-stranded

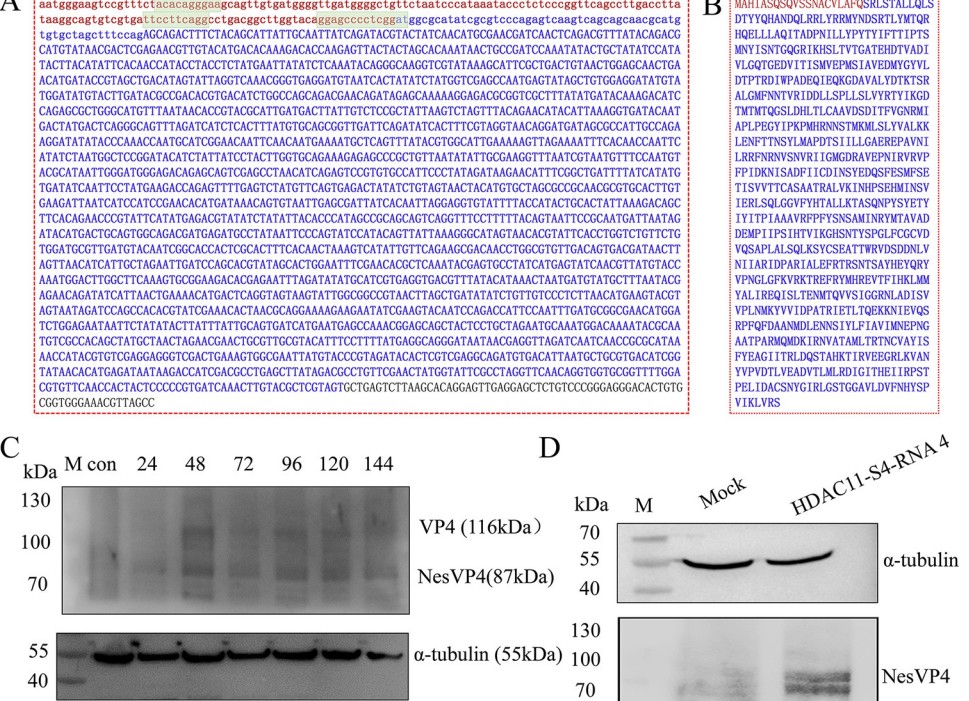

**Fig 7. Chimeric HDAC11-S4 RNA 4 encodes a truncated viral VP4 with N-terminal extensions derived from host HDAC11 RNA. A**, Chimeric HDAC11-S4 RNA 4 sequence. Lowercase letters represent the HDAC11 gene mRNA fragment with HDAC11-S4 RNA 4. Uppercase letters represent the BmCPV S4 RNA fragment with HDAC11-S4 RNA 4. Blue letters represent the speculated open reading frame, and the box represents the hypothesized IRES-like sequence. **B**, Deduced sequence of the NesVP4 protein encoded by chimeric HDAC11-S4 RNA 4. Red letters represent the amino acid sequence derived from HDAC11 mRNA, and blue letters represent the amino acid sequence derived from sense RNA of the BmCPV S4 segment. **C**, Identification of NesVP4 encoded by chimeric HDAC11-S4 RNA in the BmCPV-infected midgut by Western blotting. Proteins from the midguts of silkworms infected with BmCPV at 24–144 h postinfection were separated by SDS–PAGE. After the proteins on the gel were transferred to a nitrocellulose membrane, Western blotting was performed. An anti-VP4 antibody was used as a primary antibody, and HRP-conjugated goat anti-mouse IgG was used as a secondary antibody. α-tubulin was used as an internal reference. Lane M, molecular marker; lane con, midgut uninfected with BmCPV; lane 24-lan144, midgut infected with BmCPV at 24–144 h postinfection with BmCPV. **D**, Identification of NesVP4 in BmN cells transfected with the chimeric HDAC11-S4 RNA 4 by Western blotting. Lane M, molecular marker; lane Mock, cells without transfection; lane HDAC11-S4 RNA 4, BmN cells transfected with HDAC11-S4 RNA 4 at 48 h.

DNA viruses) [4,8,12,16,34]. In this study, chimeric silkworm-BmCPV RNAs were identified in the BmCPV-infected midgut. BmCPV is a segmented dsRNA virus belonging to the *Cypovirus* genus of the Reoviridae family [35]. To our knowledge, this is the first report showing that chimeric silkworm-BmCPV RNAs can be formed during BmCPV infection.

Previous studies have shown that in cells infected with DNA viruses that integrate into the host genome (e.g., hepatitis B virus or human papillomavirus), chimeric RNAs that are partly mapped to host RNA and partly mapped to viral RNA can be formed because the host genomic DNA region integrates viral DNA and then is transcribed [36,37,38]. BmCPV is a dsRNA virus that replicates in the cytoplasm and does not have a nuclear phase in its life cycle. To date, no event in which viral cDNA is integrated into the silkworm genome has been identified. Therefore, the chimeric silkworm-BmCPV RNAs found in this study were not produced by the integration of cDNAs originating from BmCPV RNAs into the silkworm genome and subsequent transcription.

In the process of sNSV infection, chimeric host–virus RNAs with highly diverse host-derived 5' sequences can be generated by a cap-snatching mechanism, and the RdRp encoded

by sNSVs is required for the formation of chimeric RNAs [8,9,39]. sNSVs perform cap-snatching in the nucleus, while bunyaviruses perform cap-snatching in the cytoplasm. The bunyavirus L protein is suggested to be responsible for cap binding and cleavage of host mRNA to generate a capped RNA fragment. Subsequently, the capped RNA fragment is used as a primer for viral transcription to form a chimeric mRNA [9]. In this study, we found that the fragments derived from silkworm RNA could fuse with the 5' or 3' terminus of the viral RNA fragment and that the RdRp encoded by BmCPV was not required for the formation of HDAC11-S4 RNA 4. Therefore, we hypothesized that chimeric silkworm-BmCPV RNAs were formed via a mechanism different from the cap-snatching mechanism.

It has been reported that some chimeric mRNAs can be generated by readthrough transcription and aberrant splicing during HSV-1 [15] and HIV [16] infection, respectively. The BmCPV genome consists of 10 segmented dsRNAs, and each segmented dsRNA forms a transcript, suggesting that the chimeric virus–virus RNAs found in this study were not formed by readthrough transcription.

Cis-splicing is a splicing reaction that removes introns and joins the exons included within the same RNA transcript in the nucleus to form mature RNA. Trans-splicing is a splicing reaction between two RNA molecules. Previous studies have shown that chimeric RNAs, including hybrid molecules between viral RNA and host RNA or viral RNA and viral RNA, can be generated by trans-splicing during infection by ADV, HIV, SV40 and JCV [2,3,4,6]. Usually, the sequences of the splice sites carried by RNA molecules must be located close to the consensus splice site sequences (for 5′ splice site: GU; for 3′ splice site: AG), and partial sequences in the introns of one RNA molecule complement intron regions of another RNA molecule [1,40]. CPV replication and gene transcription occur in the cytoplasm, and consensus splice site sequences matching the trans-splicing mechanism for the formation of chimeric silkworm-BmCPV RNAs were not found in the corresponding viral RNAs and silkworm RNAs. Whether chimeric BmCPV/silkworm-BmCPV RNAs are formed via a trans-splicing mechanism is worthy of further study. Moreover, it has been indicated that nonhomologous recombination and homologous recombination can occur between different RNA molecules [41]. A high frequency of homologous recombination is observed by some RNA viruses in a replicative template switch mechanism. RNA–RNA recombination has been found between viral RNAs and even between viral RNAs and host RNAs [42,43,44]. Therefore, we cannot exclude the possibility that chimeric BmCPV/silkworm-BmCPV RNAs are generated by RNA recombination.

It has been reported that host–virus chimeric RNAs identified by the RNA sequencing of cells infected with severe acute respiratory syndrome coronavirus 2 are likely artifacts arising from random template switching of reverse transcriptase and/or sequence alignment errors [45]. To eliminate misjudgment caused by sequencing artifacts, reverse transcriptional noise, template-switching and scrambled junctions of RNAs, chimeric HDAC11-S4 RNA 4 was selected for further validation. The authenticity of chimeric HDAC11-S4 RNA 4 was further confirmed by Northern blotting and cell *in situ* hybridization.

It has been indicated that some chimeric host–virus mRNAs can be translated into chimeric proteins, viral proteins with host-encoded extensions or new host–virus-encoded proteins using uAUGs within cap-snatched host transcripts as start codons [8]. In this study, two specific signal bands, one representing viral VP4 (116 kDa) and the other representing a novel protein (termed NesVP4) consisting of a truncated viral VP4 with N-terminal extensions (20 amino acid residues) derived from host HDAC11 RNA, suggesting that chimeric HDAC11-S4 RNA 4 can be translated into a protein using an AUG codon within the host RNA sequence of chimeric HDAC11-S4 RNA 4 as the start codon. BmCPV VP4, which shows RNA guanylyltransferase activity, is a viral turret protein encoded by the BmCPV S4 segment [20], and 75

amino acid residues at the N-terminus of VP4 are required for embedding viral particles into polyhedrons [46]. During virus assembly, the possibility that VP4 could be replaced by NesVP4 encoded by chimeric HDAC11-S4 RNA to form a defective virus is worthy of further study.

It has been reported that the host translational machinery is recruited to chimeric host–virus mRNAs generated via a cap-snatching mechanism, which results in global host transcriptional suppression and evasion of antiviral surveillance by RIG-I-like receptors [47]. Novel proteins translated by chimeric host–virus mRNAs impact viral replication and viral virulence and drive host T-cell immunity [14,48]. In this study, we found that viral gene expression decreased, H3K9me3 marks decreased and H3K9ac marks increased in cells transfected with chimeric HDAC-VP4 RNA 4, suggesting that viral replication and host gene expression can be regulated by chimeric HDAC-VP4 RNA 4. In short, our results indicated that during BmCPV infection, a novel mechanism different from that described in previous reports allows the genesis of chimeric silkworm-BmCPV RNAs with biological functions.

## 4 Materials and methods

### 4.1 RNA_seq

Newly molted 5th-instar silkworm larvae (Jingsong Strain) were fed mulberry leaves coated with $10^8$ BmCPV polyhedra/ml for 8 h, followed by feeding with fresh leaves at 25˚C. The midguts of silkworms infected with BmCPV were dissected at 48, 96 and 144 h postinoculation.

Total RNA was isolated from collected midguts with RNeasy R Plus Mini Kits (Qiagen, Valencia, CA, USA). After genomic DNA was digested by RNase-free DNase (Qiagen, Valencia, CA, USA), the quantity of RNA was determined with a NanoDrop 2000 Spectrophotometer (Thermo Scientific, Wilmington, USA). RNA integrity was evaluated with an Agilent 2100 Bioanalyzer (Agilent Technologies, Palo Alto, CA, USA), and RNA integrity numbers (RINs) were determined based on agarose gel electrophoresis. After ribosomal RNA was removed with an Epicenter Ribo-zero kit (Epicenter, Charlotte, NC, USA), the RNA was fragmented into 200–300 nt fragments with fragmentation buffer (New England Biol, Peking, China), followed by reverse transcription with random hexamers to generate the first strand of cDNA. Finally, cDNA libraries were constructed using RT-PCR.

The library was quantified with a Qubit 2.0 Fluorometer and diluted to 1.5 ng/μL. The quality of the cDNA library was assessed in an Agilent 2100 Bioanalyzer. Furthermore, the effective concentration was determined by quantitative PCR. The complete library was sequenced by Novogen. (Peking, China) on a HiSeq 2500 Sequencer (Illumina, San Diego, CA, USA). All sequencing data were deposited in the NCBI database under accession numbers SRR22891215, SRR22891214 and SRR22891213.

### 4.2 Preprocessing of RNA_Seq data

Illumina Casava 1.8 base-calling software was used to convert the image data obtained by the high-throughput sequencer into sequence data. The FastQC toolkit (http://www.bioinformatics.babraham.ac.uk/projects/fastqc/) was used to evaluate the quality of raw reads according to the QPhred score. After removing sequences related to adaptors with SeqPrep software (https://github.com/jstjohn/SeqPrep), low-quality reads (QPhred score < 20) were trimmed, and reads that were shorter than 50 bp and or contained N bases (unknown bases) were removed using Sickle software ([). rRNA reads were removed by aligning reads to the SILVA SSU (16S/18S) and SILVA LSU (23S/28S) databases using SortMeRNA software (http://bioinfo. lifl.fr/RNA/sortmerna/) to generate high-quality reads.

### 4.3 De novo assembly and prediction of chimeric silkworm-viral RNA

StringTie software (1.2.0) [30] was applied to assemble the obtained effective reads to generate transcripts using the *B. mori* genome (https://silkdb.bioinfotoolkits.net/resource/Bombyx_mori/download/chromosome.fa.tar.gz.) and BmCPV genomic dsRNAs (GU323605, GQ924586, GQ924587, GU323606, GQ294468, GQ294469, GQ150538, GQ150539, GQ924588, and GQ924589) as references. Subsequently, STAR-Fusion was used to analyze fusion transcripts [31], and the obtained fusion transcripts were visualized by generating a Circos diagram with R software.

### 4.4 RT–PCR and Sanger sequencing

Total RNA was isolated from collected BmCPV-infected midguts or BmN cells with RNeasy R Plus Mini Kits (Qiagen, Valencia, CA, USA). After genomic DNA was digested by RNase-free DNase (Qiagen, Valencia, CA, USA), the RNA was reverse transcribed into cDNA. Furthermore, RT-PCR was conducted with primers (S2 Table) designed according to the flanking sequences of the junction sites of predicted chimeric silkworm-viral RNAs. The PCR products were cloned into pMD-18T for sequencing.

### 4.5 Northern blotting

To further confirm the authenticity of chimeric HDAC11-S4 RNA 4, the total RNAs extracted from midguts infected with BmCPV were identified by Northern blotting using a commercial kit (Ambion, Austin, TX, USA). Briefly, total RNA (50 μg) was separated on 1% agarose–formaldehyde gels and transferred to Hybond-N+ nylon membranes (Roche, Basel, Switzerland). Northern blotting was conducted with biotin-labeled DNA (bio-CTGTAGAAAGTCTGCTG ATCGATACCGCGACG, synthesized by Sangon Biotech, Shanghai, China) targeting the junction site of chimeric HDAC11-S4 RNA 4. Signals were visualized with a Biotin Chromogenic Detection Kit (Thermo Scientific, Waltham, MA, USA). To confirm that the probe is binding to the junction site of chimeric HDAC11-S4 RNA 4 and not to HDAC11 RNA/virus S4 RNA, the prepared HDAC11-S4 RNA 4, partial HDAC11 RNA, and S4 RNA by in vitro transcription were also used for Northern blotting.

### 4.6 *In situ* hybridization

A total of $1\times10^4$ cultured BmN cells infected with BmCPV in 24-well plates at 48 h postinfection were digested with proteinase K and fixed in 4% formaldehyde at 4˚C for 1 h. The cells were hybridized with the biotin-labeled DNA probe mentioned above at 37˚C overnight according to the manual of the Ribo Fluorescent *in situ* Hybridization Kit (China, BOSTER, Cat: MK1030). CY3-labeled streptavidin was used to visualize hybridization signals. Nuclei were stained with 4,6-diamidino-2-phenylindole (DAPI). Cell images were captured using a Leica DM2000 microscope (Leica, Wetzlar, Germany). Moreover, the transfected Ctenopharyngodon idellus kidney (CIK) cells with HDAC11-S4 RNA 4, partial HDAC11 RNA, and S4 RNA, respectively, were also used for in situ Hybridization.

### 4.7 Cell transfection

A total of $1\times10^6$ cultured BmN cells in 6-well plates were transfected with pIZT-CS4 (2 μg) containing the complete cDNA sequence of BmCPV S4 dsRNA [33] or a mixture of pIZT-CS4 (1 μg) and pIZT-CS2 (1 μg) containing the complete cDNA sequence of BmCPV S2 dsRNA encoding RdRp using Roche-X Gem (Switzerland, Roche, Cat: 6366236001). At 48 h posttransfection, the cells were collected for the extraction of total RNA.

## 4.8 Characterization of the expression pattern of chimeric HDAC11-S4 RNA 4

To characterize the expression pattern of the chimeric HDAC11-S4 RNA 4, the total RNAs extracted from midguts infected with BmCPV at 0, 24, 48, 72, 96, 120 and 144 h postinoculation were reverse transcribed into cDNA using random primers (First Strand cDNA Synthesis Kit, Transgene, Beijing, China). The expression levels of the chimeric HDAC11-S4 RNA 4 were determined by qRT-PCR using a pair of specific primers, qHCPV20-4 (S2 Table). The translation initiation factor eIF-4A (TIF-4A) gene was utilized as an internal reference.

## 4.9 Plasmid construction and transcription *in vitro*

The cDNA sequence of chimeric HDAC11-S4 RNA 4 (Fig 7A) controlled by the T7 promoter was chemically synthesized and cloned into the *Eco*RI and *Hin*dIII sites of the pUC57 vector to generate pUCHCPV20-4. After being digested by *Hin*dIII, linearized pUCHCPV20-4 was used as a template for *in vitro* transcription by T7 RNA polymerase (TaKaRa, Dalian, China) to generate chimeric HDAC11-S4 RNA 4. After removing the plasmid DNA with DNaseI, the chimeric HDAC11-S4 RNA 4 was purified by extraction with phenol/chloroform treatment and precipitation with ethanol. A similar method was used to generate CPV S4 RNA. The partial HDAC11 RNA was amplified by T7HDAC11 primers (S2 Table) using pUCHCPV20-4 as template. After amplification, the partial HDAC11 DNA was used as a template for *in vitro* transcription by T7 RNA polymerase to generate partial HDAC11 RNA. A similar method was used to generate GFP RNA.

## 4.10 Effect of chimeric HDAC11-S4 RNA 4 on BmCPV viral gene expression and histone modification

First, $1 \times 10^6$ BmN cells (1 ml) were transfected with chimeric HDAC11-S4 RNA 4 (1, 2 or 3 μg), and 24 h later, the cells were inoculated with BmCPV (MOI = 2). The collected cells at 48 h postinoculation were used for detection of viral gene expression and modification of histone 3. For viral gene expression, the transcription level of the *vp1* gene was determined by qRT–PCR with CPV-S1 primers (S2 Table). The levels of VP7, H3K9me3 and H3K9ac were determined by Western blotting with corresponding antibodies. Histone 3 was used as an internal reference. Cells transfected with the corresponding doses of GFP RNA, partial HDAC11 RNA, CPV S4 RNA were used as controls, respectively.

## Supporting information

**S1 Fig. The flanking sequences of the breakpoints of left/right parental RNAs.**
(TIF)

**S2 Fig. The flanking sequence of the junction site of HDAC11-S4 RNA 4.**
(TIF)

**S3 Fig. Northern and In situ hybridization to confirm the probe is binding the junction site of chimeric HDAC11-S4 RNA 4. A, Northern blotting for HDAC11-S4 RNA 4, partial HDAC11 RNA, and CPV S4 RNA.** HDAC11-S4 RNA 4, partial HDAC11 RNA, and CPV S4 RNA prepared by in vitro transcription were separated on 1% agarose-formaldehyde gels and transferred to Hybond-N+ nylon membranes. Northern blotting was conducted with biotin-labeled DNA targeting the junction site of chimeric HDAC11-S4 RNA 4. **B, In situ hybridization of *Ctenopharyngodon idellus* kidney (CIK) cells respectively transfected with HDAC11-S4 RNA 4, partial HDAC11 RNA, and CPV S4 RNA.** A total of $1 \times 10^4$ cultured

CIK cells seeded in 24-well plates, and followed by transfection with partial HDAC RNA (2μg), BmCPV S4 RNA (2μg) or HDAC11-S4 RNA 4 (2μg). The cells were collected at 48 h post transfection and digested with proteinase K and fixed in 4% formaldehyde at 4˚C for 1 h. The cells were hybridized with the biotin-labeled DNA targeting the junction site of chimeric HDAC11-S4 RNA 4. CY3-labeled streptavidin was used to visualize hybridization signals. Nuclei were stained with DAPI.
(TIF)

**S1 Table. Information for chimeric silkworm-BmCPV RNAs identified in the midguts of silkworms infected with BmCPV.**
(PDF)

**S2 Table. The primers used in this study.**
(PDF)

**S3 Table. Data used to draw the conclusions.**
(XLSX)

## Author Contributions

**Conceptualization:** Chengliang Gong.

**Data curation:** Jun Pan, Shulin Wei.

**Formal analysis:** Jun Pan.

**Funding acquisition:** Xiaolong Hu, Chengliang Gong.

**Investigation:** Shulin Wei, Qunnan Qiu, Xinyu Tong, Zeen Shen.

**Methodology:** Jun Pan, Zeen Shen.

**Project administration:** Min Zhu.

**Resources:** Xiaolong Hu, Chengliang Gong.

**Supervision:** Min Zhu, Xiaolong Hu.

**Validation:** Shulin Wei, Qunnan Qiu, Xinyu Tong.

**Visualization:** Shulin Wei.

**Writing – original draft:** Jun Pan, Chengliang Gong.

**Writing – review & editing:** Shulin Wei, Chengliang Gong.

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
