## [Decision Letter · Decision Letter 0]

19 Apr 2023

Dear Dr. Gong,

Thank you very much for submitting your manuscript "A novel chimeric RNA originating from BmCPV S4 and Bombyx mori HDAC11 transcripts regulates virus proliferation" for consideration at PLOS Pathogens. As with all papers reviewed by the journal, your manuscript was reviewed by members of the editorial board and by several independent reviewers. In light of the reviews (below this email), we would like to invite the resubmission of a significantly-revised version that takes into account the reviewers' comments.

We cannot make any decision about publication until we have seen the revised manuscript and your response to the reviewers' comments. Your revised manuscript is also likely to be sent to reviewers for further evaluation.

Sincerely,

Laura B. Goodman, PhD

Guest Editor

PLOS Pathogens

Sara Cherry

Section Editor

PLOS Pathogens

Kasturi Haldar

Editor-in-Chief

PLOS Pathogens

orcid.org/0000-0001-5065-158X

Michael Malim

Editor-in-Chief

PLOS Pathogens

orcid.org/0000-0002-7699-2064

Thank you for your patience in the review process. We have obtained the following reviews, which you are invited to consider and resubmit your manuscript after making the indicated changes. Please pay particular attention to the concerns raised about the appropriateness of controls and the validation of the probes for several of the experiments.

Reviewer's Responses to Questions

**Part I - Summary**

Reviewer #1: Pan et al. have discovered several novel chimeric RNAs whereby the silkworm RNA has fused with the BmCPV virus. It has previously been reported that start codons with cap-snatched host transcripts or trans-splicing can generate chimeric RNAs on ssRNA and dsDNA viruses; however, whether chimeric RNAs can be generated on dsRNA viruses remains a mystery. Here the authors use BmCPV (a dsRNA virus) infection of the silkworm to demonstrate the presence of several host-virus chimeras including HDAC11-S4 RNA4, derived from the silkworm histone deacetylase 11 and the BmCPV S4 transcript encoding viral structural protein 4 (VP4).

They validated the presence of the RNA chimera using northern blot and in situ hybridization, then showed that the transcript could be translated into a truncated BmCPV CP4 with a silkworm HDAC11-derived N-terminal extension. Further, they found that the chimeric RNA inhibited virus proliferation and decreased histone methylation, and increased histone acetylation.

Reviewer #2: The manuscript titled ‘A novel chimeric RNA originating from BmCPV S4 and Bombyx mori HDAC11 transcripts regulates proliferation’ by Pan et al. show the presence of host-viral chimeric RNA in the midgut of BmCPV-infected silkworms. Using RNA_seq and Sanger sequencing they observed the chimera of host histone deacetylase 11 and BmCPV S4 RNA HDAC11-S4 RNA in BmN cells transfected with cDNA of BmCPV S4 both in presence or absence of S2 cDNA encoding RNA dependent RNA polymerase (RdRp), denoting that the chimeric RNA was not mediated by RdRp. The authors make the point that this is the first instance that a RdRp independent chimeric RNA mechanism of synthesis is being observed in silkworm-infected with virus. They further demonstrate the characteristics/fate of the chimeric RNA using qPCR that the expression increases upon infection but decreases 140 hrs after infection, finally showing that these RNAs cannot be packaged into virions. By monitoring the expression of other viral genes like vp1 and protein expression of VP7, they found that the expression of viral genes were decreased when infected with these chimeric RNAs., suggesting that the chimeric RNA causes inhibition of BmCPV replication. They then observe using Western blot that this chimeric HDAC11-S4 RNA decreased trimethylation of histone 3 lysine 9 but increased the histone 3 lysine 9 acetylation, highlighting a possible biological effect of such chimeric RNA on host cells. Western blotting detection using anti-VP4 antibody in the BmCPV infected midgut cells showed not only the presence of viral protein VP4 but also a truncated NesVP4 protein product translated from another ORF starting at the N-terminal extension derived from the host RNA. NesVP4 protein product was also detected when in vitro transcribed HDAC11-S4 mRNA was transfected into the BmN cells.

Overall the work is sound on technique and establishes the presence and effect of host-viral chimeric RNA in silkworms upon BmnCPV viral infection. The only issue for this reviewer is the way the text has been presented in the introduction misleads the readers to think that this work is based on yet another host-viral cap-snatching mechanism of chimeric RNA formation. On that note, the text highly resembles that of previously published papers on cap-snatching. While the introduction has tried to summarize all the diverse examples of chimeric host-viral chimeric RNA origins, it fails to lead into the main take-home of this work, which is the identification of a distinct mechanism of chimeric RNA in silkworm-infected with BmnCPV. Although chimeric RNA formation using a cap-snatching mechanism has been previously reported for many different host-virus infections, this work shows and characterizes for the first time the occurrence of silkworm-BmCPV chimeric RNA possibly generated using a different mechanism, further attempting to characterize the biological effect of this chimeric RNA in virus-infected BmN cells.

Reviewer #3: Pan et al. identified host-virus chimeric RNAs in a dsRNA virus infection model (BmCPV in silkworm), which is a novel phenomenon. They do follow-up experiments with one specific chimeric RNA and suggest it may be functionally active and biologically relevant.

The key experiments do demonstrate that the chimeric protein has activity but fall somewhat short of proving that the chimeric protein provides novel functionality compared to either of the normal versions of the two components on their own.

**Part II – Major Issues: Key Experiments Required for Acceptance**

Reviewer #1: For Figure 4, since the probe targets the junction (i.e., ½ of the probe is binding to the host gene), I am surprised that no signal is observed from the control sample with no infection (I would have expected some binding to this region). Need evidence to show that the probe is binding the junction and not BmCPV on its own.

I have the same concern about the FISH probe in figure 4. Need evidence to show that the probe is binding to the junction and not just the BmCPV

Reviewer #2: (No Response)

Reviewer #3: • Lines 222-224: No data are shown to support the claim that the chimeric HDAC11-S4 RNA 4 is not packaged into virions.

• Figure 6 and related text:

o What was the rationale for choosing chimeric HDAC11-S4 RNA 4 for further workup? Was it the most abundant? Or was it chosen randomly?

o Why did the authors not choose wild-type HDAC11 and/or the short N-terminal fragment found in the chimeric protein and/or wild-type VP4 as a control for this experiment, and also transfect that in increasing doses? This would not only be a better control in general (see my next point), it would also let them address whether this chimeric protein offers any additional functionality compared to the normal cellular HDAC11 and the normal viral VP4.

o To demonstrate that the changes in vp1 RNA, VP7 protein, H3K9me3 and H3K9ac are all specific effects mediated by increasing dosages of HDAC11-S4 RNA 4, all experiments shown in this figure should be repeated with matching increasing dosages of a control RNA. Using only a single dose of GFP RNA doesn’t rule out non-specific effects.

**Part III – Minor Issues: Editorial and Data Presentation Modifications**

Reviewer #1: 1. Sometimes RT-PCR is referred to as PCR in the text

2. It would be helpful if the sequences for BMSK_chr11 were aligned in Figure 2a and 2b (they are currently offset by a few nucleotides) and a reference sequence for BMSK_chr11 the viral inserts were aligned there as well.

3. Same comment above for the other examples in figure 2 and figure 3, reference sequences would be helpful to see sequence similarities, etc.

4. Did all instances of the host transcripts found in chimera form become this way, or is there still a fraction of these transcripts that does not become a chimera? Can the authors comment on or estimate the percentages?

5. For figure 4, the full gel, including the size markers, should be shown rather than a cutout.

6. In the methods section where the FISH experiment is described, there is a typo–streptomycin should be cy3-labeled streptavidin, and the images in figure 4 should have scale bars.

7. Axis labels for figure 5

Reviewer #2: Minor points:

Line 268 - Rephrase for grammar error, maybe like - ‘because the host genomic DNA region integrates viral DNA and then is transcribed’

Reviewer #3: • Figure 6 and related text:

o If the authors want to suggest that the chimeric protein encoded by HDAC11-S4 RNA 4 is potentially biologically relevant, they should comment on its abundance. If the RNA is vanishingly rare, it will likely not matter in vivo. Their NGS data should enable the authors to discuss this point.

o Lines 189-192: RNA 4 also does not match the reference sequence (see fig.3D). It has a 1nt mismatch (T895A). Do the authors think this is an artifact? Would this affect the amino acid sequence?

• Figure legends are missing important information, for example:

o Figure 1A: What do the red and blue lines in the centers of the Circos plots indicate?

o Figure 1A: The legend should say that “BMSK_x” indicates the individual silkworm chromosomes.

o Figure 1B: What are the circles at one end of the wavy lines representing transcripts?

o Figures 5 and 6: What is the basis for the error bars? How many replicates are shown? Biological replicates or technical replicates? Is any of it significant?

PLOS authors have the option to publish the peer review history of their article (what does this mean?). If published, this will include your full peer review and any attached files.

Reviewer #1: No

Reviewer #2: No

Reviewer #3: No
---

## [Decision Letter · Decision Letter 1]

17 Nov 2023

Dear Dr. Gong,

We are pleased to inform you that your manuscript 'A novel chimeric RNA originating from BmCPV S4 and Bombyx mori HDAC11 transcripts regulates virus proliferation' has been provisionally accepted for publication in PLOS Pathogens.

Best regards,

Sara Cherry

Section Editor

PLOS Pathogens

Sara Cherry

Section Editor

PLOS Pathogens

Kasturi Haldar

Editor-in-Chief

PLOS Pathogens

orcid.org/0000-0001-5065-158X

Michael Malim

Editor-in-Chief

PLOS Pathogens

orcid.org/0000-0002-7699-2064

Reviewer Comments (if any, and for reference):

Reviewer's Responses to Questions

**Part I - Summary**

Reviewer #1: The authors have addressed all of the points from the reviewers

Reviewer #2: The authors have addressed my comments.

**Part II – Major Issues: Key Experiments Required for Acceptance**

Reviewer #1: The authors have addressed all of the points from the reviewers

Reviewer #2: (No Response)

**Part III – Minor Issues: Editorial and Data Presentation Modifications**

Reviewer #1: (No Response)

Reviewer #2: (No Response)

PLOS authors have the option to publish the peer review history of their article (what does this mean?). If published, this will include your full peer review and any attached files.

Reviewer #1: No

Reviewer #2: No

---

## [Editor Report · Acceptance letter]

28 Nov 2023

Dear Dr. Gong,

We are delighted to inform you that your manuscript, "A novel chimeric RNA originating from BmCPV S4 and *Bombyx mori* HDAC11 transcripts regulates virus proliferation," has been formally accepted for publication in PLOS Pathogens.

Best regards,

Kasturi Haldar

Editor-in-Chief

PLOS Pathogens

orcid.org/0000-0001-5065-158X

Michael Malim

Editor-in-Chief

PLOS Pathogens

orcid.org/0000-0002-7699-2064